# High Rates of Antimicrobial Resistance in Rapidly Growing Mycobacterial Infections in Taiwan

**DOI:** 10.3390/pathogens11090969

**Published:** 2022-08-25

**Authors:** Hui-Zin Tu, Herng-Sheng Lee, Yao-Shen Chen, Susan Shin-Jung Lee

**Affiliations:** 1Division of Microbiology, Department of Pathology and Laboratory Medicine, Kaohsiung Veterans General Hospital, Kaohsiung 813, Taiwan; 2Department of Pathology and Laboratory Medicine, Kaohsiung Veterans General Hospital, Kaohsiung 813, Taiwan; 3Department of Administration, Kaohsiung Veterans General Hospital, Kaohsiung 813, Taiwan; 4Division of Infectious Diseases, Department of Internal Medicine, Kaohsiung Veterans General Hospital, Kaohsiung 813, Taiwan; 5Faculty of Medicine, School of Medicine, National Yang Ming Chiao Tung University, Taipei 112304, Taiwan; 6School of Medicine, College of Medicine, National Sun Yat-sen University, Kaohsiung 804, Taiwan

**Keywords:** rapidly growing mycobacteria, minimum inhibitory concentration, broth microdilution test, E-test

## Abstract

Rapidly growing mycobacteria (RGM) has gained increasing clinical importance, and treatment is challenging due to diverse drug resistance. The minimum inhibitory concentrations (MIC) of 13 antimicrobial agents using modified broth microdilution and E-test were determined for 32 clinical isolates of RGM, including *Mycobacterium abscessus* (22 isolates) and *Mycobacterium fortuitum* (10 isolates). Our results showed high rates of resistance to available antimicrobial agents. Amikacin remained highly susceptible (87.5%). Clarithromycin was active against the isolates of *M. abscessus* (95.5%), and *M. fortuitum* (50%), but 36.4% and 20% had inducible macrolide resistance, respectively. Rates of susceptibility to tigecycline were 68.2–70%, and linezolid 45.5–50%, respectively. The quinolones (ciprofloxacin and moxifloxacin) showed better in vitro activity against *M. fortuitum* isolates (50% susceptibility) than the *M. abscessus* isolates (31.8% susceptibility). The susceptibilities to other conventional anti-mycobacterial agents were poor. The MICs of E-test were higher than broth microdilution and may result in reports of false resistance. In conclusion, the implementation of the modified broth microdilution plates into the routine clinical laboratory workflow to provide antimicrobial susceptibility early, allows for the timely selection of appropriate treatment of RGM infections to improve outcome.

## 1. Introduction

Non-tuberculous mycobacteria (NTM) are increasingly prevalent human pathogens, ubiquitously found in the natural environment, such as water, soil and food [1,2,3]. Recently, the prevalence rates of NTM have increased in many countries with marked geographical differences in the species encountered, including Taiwan [4,5,6,7,8,9,10,11]. The proportion of NTM contributed by rapidly growing mycobacteria (RGM) has increased more than 2-fold from 14% during the period of 1992–1996, to 35% in 2001 in Taiwan [9,10]. Among the RGM, *M. abscessus*, *M. fortuitum* and *M. chelonae* accounted for the majority of both community-acquired and health care-associated diseases [12]. Infections involved mostly the skin and soft tissue infection, but also, pulmonary, skeletal and disseminated infections [4,12,13].

The RGM are known to be resistant to conventional anti-tuberculous drugs and susceptibilities vary widely with different mycobacterial species [14,15,16]. The treatment strategy of RGM is based on in vitro antimicrobial susceptibility testing. Broth microdulion and E-test are two methods often used in antimicrobial susceptibility testing. Broth microdilution is the most commonly used method and provides both quantitative (minimal inhibition concentration, MIC) and qualitative (category interpretation) results. However, conventional broth microdilution method based on the guidelines of the Clinical Laboratory and Standards Institute (CLSI) is time-consuming, labor-intensive, and difficult to interpret [17,18]. Some commercial microdilution panels (e.g., Sensititre RAPMYCOI panel) are available for the testing of RGM against amikacin, cefoxitin, ciprofloxacin, clarithromycin, doxycycline, imipenem, linezolid, sulfamethoxazole, tobramycin and tigecycline [19,20,21]. Another alternative is the E-test, which provides a quantification of antimicrobial susceptibility of microorganisms by placing an antibiotic impregnated strip with exponential concentration gradient on the agar plate. This is a quantitative method that applies both the dilution of antibiotics and diffusion of antibiotics into the medium. MIC results by E-test showed higher antimicrobial resistance than those by the golden standard [22,23].

Macrolides are a main component in the treatment of RGM infections. Inducible macrolide resistance occurs in some subspecies with a functional erythromycin ribosome methyltransferase (*erm*) gene, which encodes an rRNA methyltransferase, that modifies the binding site for macrolides. The CLSI recommend testing for inducible macrolide resistance which can be tested by prolonged incubation of microdilution tray or by *erm* gene mutation detection [24].

This study aims to compare the antimicrobial susceptibility of RGM using the broth microdilution test and E-test for in vitro susceptibility testing, and to determine the optimal testing method feasible for routine use in clinical laboratories. The outcome of this study may assist the clinicians to choose the most appropriate antimicrobial therapy based on susceptibility results to improve treatment outcome.

## 2. Materials and Methods

### 2.1. Study Isolates

Based on compatible clinical symptoms and/or histopathological findings, we selected 32 RGM isolates obtained from sterile sites by aseptic techniques from patients. These 32 sterile sites included 11 pleural effusion, 8 biopsy tissue, 8 wound pus, 3 blood, 1 synovial fluid, and 1 vitreous fluid. These 32 isolates were identified as *M. abscessus* isolates (n = 22) and *M. fortuitum* isolates (n = 10) by PCR restriction fragment length polymorphism analysis (PRA) [25]. Isolates were stored at −80 °C and re-cultured to the Middlebrook 7H11 agar at 35 °C. Subcultures to TSA (trypticase soy agar) were performed on the Middlebrook 7H11 agar in order to obtain pure colonies for broth microdilution tests and E-test.

### 2.2. Broth Microdilution Test

The Sensititre RAPMYCOI plate (Trek Diagnostic Systems Ltd., East Grinstead, UK) with 2-fold diluted drugs in the 96 wells was prepared [26,27,28]. Antimicrobial susceptibilities of the RGM strains against 15 anti-tuberculous drugs, including the MIC, MIC_50,_ and MIC_90_ as well as the susceptible proportions were determined. MIC is the lowest concentration of an antibiotic that prevents the visible growth of bacteria. MIC_50_ and MIC_90_ values were defined as the lowest concentration of the antibiotic at which 50 and 90% of the isolates were inhibited, respectively. Inoculum suspensions were prepared in sterile water to a density of 0.5 MacFarland standards. Fifty µL of suspension were transferred to a tube of cation adjusted Mueller-Hinton broth with TES buffer (CAMHBT) (Trek Diagnostic Systems Ltd.). One hundred µL of this mixture was transferred to each well of the Sensititre CAMHBT plate containing antibiotics in appropriate dilutions. All the wells were covered with adhesive seal and incubated at 30 °C in a non-CO_2_ incubator for 3–5 days. The interpretation of drug susceptibility tests as susceptible, intermediate, and resistant adhered to the CLSI M24-A2. Extended 14 days incubation for clarithromycin susceptibility testing was performed and the MIC results of the clarithromycin between two different incubation periods (day 5 and day 14) were compared, including the MIC range, MIC_50_, MIC_90_ and susceptible proportions. *Staphylococcus aureus* ATCC29213 and *Mycobacterium peregrinum* ATCC700686 were used as quality control strains.

### 2.3. E-Test

E-test antibiotic strip (bioMérieux, Marcy l’Étoile, France) was used to determine MICs of RGM based on the CLSI M24-A2 [22,23,29,30]. The MIC was determined by the number marked at the junction of the bacterial growth plate and the antibiotic strip. If the value observed was less than the 2-fold diluted concentration, then the next higher concentration was recorded as its MIC. For example, if the strip read 12.5 µg/mL, then 16 µg/mL would be the appropriate MIC. *Staphylococcus aureus* ATCC29213 and *Enterococcus faecalis* ATCC29212 were used as quality control strains.

## 3. Results

### 3.1. Susceptibility Analysis by Broth Microdilution

The MIC, MIC_50_, and MIC_90_ as well as the susceptible proportions were tested on 32 RGM isolates (22 *M. abscessus* isolates and 10 *M. fortuitum* isolates) (Table 1). Most of RGMs remained susceptible to amikacin (86.4% for *M. abscessus* and 90% for *M. fortuitum*). Inhibition of RGMs by amikacin was better than that of tobramycin. *M. fortuitum* isolates showed susceptibility to ciprofloxacin (50%) and moxifloxacin (50%), but the proportion of *M. abscessus* susceptible to quinolones was only 31.8% for both drugs. Clarithromycin inhibited *M. abscessus* better than *M. fortuitum* (59.1% vs. 30%). The tested RGM demonstrated to be less susceptible (less than 25%) to cefoxitin, doxycycline, minocycline, imipenem, and tobramycin. Trimethoprim-sulfamethoxazole (SXT) had low susceptibility rates of 30%, while linezolid susceptibility rates were lower than 50%. The MIC_50_ of amoxicillin-clavulanic acid, cefepime and ceftriaxone in both species were all greater than 32 μg/mL. However, the CLSI guidelines do not provide interpretations of MIC breakpoints of amoxicillin-clavulanic acid, cefepime and ceftriaxone for RGM. The MICs of all test strains against the drug “tigecycline” were low; below 1 μg/mL. In the result of E-test, tigecycline showed better activity (68.2% for *M. abscessus* and 70% for *M. fortuitum*) than the other antimicrobial agents tested, except for amikacin [10,30,31,32].While the MIC_50_ and MIC_90_ of the *M. abscessus* were 0.25 μg/mL and 1 μg/mL, respectively; those of *M. fortuitum* were 0.25 μg/mL and 0.5 μg/mL, respectively. The comparison of proportion susceptibilities of *M. abscessus* and *M. fortuitum* by the broth microdilution test and the E-test is shown in Table 2 and Figure 1.

### 3.2. Inducible Macrolide Drug Resistance

The comparison of MIC range, MIC_50_, MIC_90_ and susceptible proportions results of clarithromycin between two different incubation periods (day 5 and day 14) were shown in Table 3. Inducible clarithromycin resistance was observed in both *M. abscessus* and *M fortuitum* strains. Susceptibility rates of *M. abscessus* decreased from 95.5% to 59.1% with an extended 14 day incubation period, while *M. fortuitum* decreased from 50% to 30%, indicating the presence of inducible macrolide resistance in 36.4% and 20% of *M. abscessus* isolates and *M. fortuitum* isolates, respectively.

### 3.3. E-Test Analysis

The E-test analysis demonstrated higher levels of antimicrobial resistance among the RGMs compared to broth microdilution in 90% of the isolates (Figure 1). The rates of resistance to amikacin were 68%, 82% for ciprofloxacin, 86% for trimethoprim-sulfamethoxazole, 91% for imipenem, linezolid, minocycline and 100% for doxycycline (Table 4). The only exception was a lower rate of resistance to clarithromycin in the *M. abscessus*. *M. fortuitum* demonstrated 30% resistance to amikacin, 40% for ciprofloxacin, and more than 70% resistance to all the other drugs. The proportions of resistant strains were 70% for clarithromycin, linezolid, minocycline, trimethoprim-sulfamethoxazole, 80% for doxycycline, and 100% for imipenem (Table 1). The comparison between antimicrobial susceptibility of RGM across different time periods in Taiwan from previous studies is shown in Table 5.

## 4. Discussions

Antimicrobial resistance of RGM is known to vary across different species, time periods, and geographical locations [12,14]. This study found that among the antimicrobial agents tested, amikacin had a high susceptibility rate in both the *M. abscessus* and *M. fortuitum.* The proportion of the *M. abscessus* susceptible to clarithromycin was higher than that of *M. fortuitum*, but susceptibility rates to quinolones were lower than in *M. fortuitum*. Other antimicrobial agents tested showed high rates of resistance for both *M. abscessus* and *M. fortuitum.* A decrease in antimicrobial susceptibility of the *M. abscessus* and *M. fortuitum* was found over time. The MICs by the E-test method were generally higher than the broth microdilution method and may result in reports of false resistance. To ensure that appropriate treatment regimens are prescribed, we recommend regular or periodic surveillance of the antimicrobial resistance among RGM, if routine susceptibility testing is not available.

In our study, the susceptibility rates to amikacin were 86% and 90% against *M. abscessus* and *M. fortuitum*, respectively. This was similar to other studies in which amikacin was reported to demonstrate more than 95% efficacy [9,10,23]. Macrolides (clarithromycin) showed 59.1% and 30% efficacy against *M. abscessus* and *M. fortuitum*, respectively, which is lower than the previous reports of 52–100% susceptibility against *M. abscessus* and 14.6–80% against *M. fortuitum* [9,10,23]. Due to the possibility of inducible macrolide resistance, the CLSI M24-A2 suggests that interpretation of antimicrobial susceptibility to clarithromycin be extended to 14 days of incubation [30,31,32,33,34]. The mechanisms may involve mutations of the *erm* gene [30,35], although further molecular typing was not conducted in this study. The *erm* gene encodes an rRNA methyltransferases, which modifies the binding site of macrolides, conferring resistance. Inducible clarithromycin resistance was demonstrated in 36.4% of *M. abscessus* and 20% of *M. fortuitum* isolated in our study. This rate of inducible macrolide resistance was lower than the 53.6% found in another recent study conducted in northern Taiwan, in *M. abscessus* subsp. *abscessus* group from skin and soft tissue infections [36].

The susceptibility rates to newer treatment options recommended by the treatment guidelines for nontuberculous mycobacteria from the British Thoracic Society (BTS) and the American Thoracic Society/Infectious Diseases Society of America (ATS/IDSA) [37,38], including linezolid and tigecycline, was performed in this study. Tigecycline is a viable option for treatment of RGM with a susceptibility rate of 68.2–70%, but caution is recommended for empirical use of linezolid, which had lower susceptibility rates of 45.5–50%. Susceptibility rates for tigecycline had geographical variation, with rates of 33~100% in *M. abscessus* and 57% for *M. fortuitum*, in a study from central Taiwan. Synergistic activity was demonstrated when tigecycline is combined with clarithromycin, against 92.9%, 68.8%, 100%, and 35.7% of *M. abscessus* subsp. *abscessus*, subsp. *massiliense*, subsp. *bolletii*, and M. *fortuitum* isolates, respectively [39]. Linezolid also demonstrated variable susceptibility rates of 32% and 68% in *M. abscessus* and *M. fortuitum*, respectively, in an earlier study from northern Taiwan [9].

In general, it has been reported that E-test has higher MIC levels of antimicrobial resistance among the RGMs compared to broth microdilution. Our findings are consistent with previous studies and showed that the MICs of E-test are higher, up to two fold, than the broth microdilution test [9]. The exact reasons for the discrepancies are not known, but differences in the performance or interpretation of the E-test may play a role, as the E-test has not been standardized for the RGM. Therefore, although the E-test has the advantage of being is recognized as an inexpensive and quick method [22,23], it may result in false resistance due to higher MICs. Although broth microdilution is the standard method recommended by the CLSI for testing the antimicrobial susceptibility of RGM, routine clinical testing using this method is unfeasible due to the labor-intensiveness. The commercialized, modified, Sensititre RAPMYCOI plates are now recommended as a choice for antimicrobial susceptibility testing of RGM [2,40,41,42]. This is a simple test that can be implemented routinely in the clinical laboratory workflow.

The limitation in this study is that we did not perform the molecular typing of the *M. abscessus*, which can be further differentiated three subspecies: *M. abscessus* subsp. *abscessus*, *M. abscessus* subsp. *massiliense*, and *M. abscessus* subsp. *bolletii.* The subsp. *massiliense* has a dysfunctional *erm*(41) gene and remains susceptible to macrolides. In the subsp. *abscessus* and subsp. *bolletii*, *genetic* polymorphism at nucleotide 28 of the *erm* gene confers resistance, with wild-type T28 sequevars demonstrating inducible clarithromycin resistance while C28 sequevars remain susceptible [35,43]. However, we performed an extended 14 day incubation to allow for detection of inducible clarithromycin resistance, which enabled appropriate antimicrobial treatment.

## 5. Conclusions

In conclusion, our study demonstrated that RGM remains susceptible to amikacin, and clarithromycin, but inducible macrolide resistance may occur in over one-third of isolates. An extended 14 day incubation is recommended by current guidelines to detect inducible macrolide resistance [37,38]. There is geographical variation in the susceptibility rates of newer treatment options, such as linezolid and tigecycline, and susceptibility testing is advised to optimize the treatment strategy. E-tests will likely result in false resistance due to the higher MICs compared to the broth microdilution tests. The implementation of the modified broth microdilution plates into the routine clinical laboratory workflow to provide antimicrobial susceptibility early, allows the timely selection of appropriate treatment of RGM infections to improve outcome.

## Figures and Tables

**Figure 1 pathogens-11-00969-f001:**
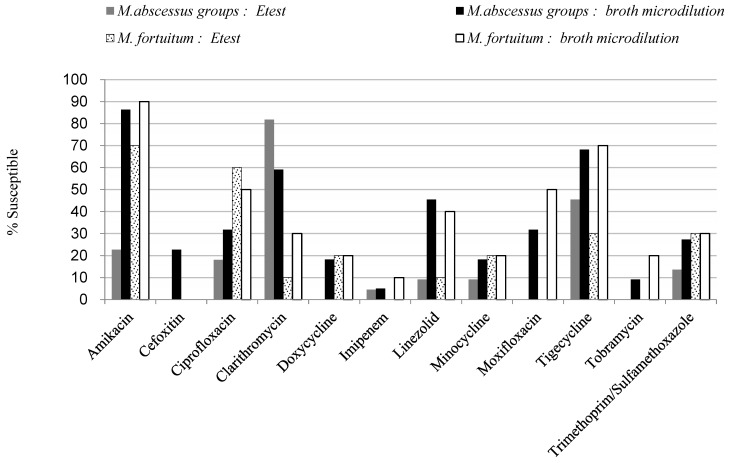
Antimicrobial susceptibility of the rapidly growing mycobacteria (RGM) by E-test and modified broth microdilution method. Compared to the modified broth microdilution method, the E-test method yielded lower rates of antimicrobial susceptibility to amikacin, ciprofloxacin, linezolid, minocycline, moxifloxacin, trimethoprim/sulfamethoxazole for *M. abscessus* group; and in contrast, the antimicrobial susceptibility rates to *M. fortuitum* were only lower for linezolid, tigecycline and moxifloxacin using the E-test method.

**Table 1 pathogens-11-00969-t001:** In vitro susceptibilities of isolates from various sterile body sites of the rapidly growing mycobacteria using the broth microdilution method.

Organism (No. of Isolates Tested) and Antimicrobial Agents	MIC (μg/mL)			No. (%) Isolates		
	Range	50%	90%	Susceptible	Intermediate	Resistant
***M. abscessus*** (22)						
Amikacin	≦1–64	8	32	19 (86)	2 (9)	1 (5)
Amoxicillin/Clavulanic acid	8->64	>64	>64	– ^§^	–	–
Cefepime	16->32	>32	>32	–	–	–
Cefoxitin	8–64	32	64	5 (23)	17 (77)	0 (0)
Ceftriaxone	16->64	>64	>64	–	–	–
Ciprofloxacin	0.5->4	4	>4	7 (32)	0 (0)	15 (68)
Clarithromycin *	0.12->16	1	>16	13 (59)	0 (0)	9 (41)
Doxycycline	0.25->16	>16	>16	4 (18)	0 (0)	18 (82)
Imipenem	2->64	16	64	1 (5)	17 (77)	4 (18)
Linezolid	≦1->32	16	>32	10 (45)	2 (9)	10 (45)
Minocycline	≦1->8	>8	>8	4 (18)	0 (0)	18 (82)
Moxifloxacin	≦0.25->8	8	>8	7 (32)	2 (9)	13 (59)
Tigecycline	0.03–1	0.12	1	–	–	–
Tobramycin	2->16	16	>16	2 (9)	5 (23)	15 (68)
Trimethoprim/Sulfamethoxazole	≦0.25->8	>8	>8	6 (27)	0 (0)	16 (73)
***M. fortuitum*** (10)						
Amikacin	≦1–32	2	32	9 (90)	1 (10)	0 (0)
Amoxicillin/Clavulanic acid	8->64	32	>64	–	–	–
Cefepime	>32	>32	>32	–	–	–
Cefoxitin	32->128	64	>128	0 (0)	9 (90)	1 (10)
Ceftriaxone	64->64	>64	>64	–	–	–
Ciprofloxacin	≦0.12->4	≦0.12	>4	5 (50)	0 (0)	5 (50)
Clarithromycin *	1->16	>16	>16	3 (30)	1 (10)	6 (60)
Doxycycline	0.25->16	>16	>16	2 (20)	0 (0)	8 (80)
Imipenem	≦2–32	16	32	1 (10)	2 (20)	7 (70)
Linezolid	2->32	4	>32	4 (50)	3 (38)	1 (13)
Minocycline	≦1->8	>8	>8	2 (20)	0 (0)	8 (80)
Moxifloxacin	≦0.25->8	≦0.25	>8	5 (50)	0 (0)	5 (50)
Tigecycline	0.06–0.5	0.25	0.5	–	–	–
Tobramycin	4->16	16	>16	2 (20)	0 (0)	8 (80)
Trimethoprim/Sulfamethoxazole	≦0.25->8	4	>8	3 (30)	0 (0)	7 (70)

* Result of Clarithromycin was analyzed after 14 days. Others drugs were 5 days. ^§^ Reference range not provided by CLSI M24-A2 guideline. – not available.

**Table 2 pathogens-11-00969-t002:** Comparison between in vitro susceptibilities of rapidly growing mycobacterial isolates from clinical samples of various sterile body sites using the E-test and the modified broth microdilution method.

Organism (no. of Isolates Tested) and Antimicrobial Agents	MIC 50% (μg/mL)		No (%) of Susceptible Isolates	
	E-Test	Microdilution	E-Test	Microdilution
***M. abscessus*** (22)				
Amikacin	>256	8	5 (22.7)	19 (86.4)
Cefoxitin		32	–	5 (22.7)
Ceftriaxone	>32	>64	– ^§^	– ^§^
Ciprofloxacin	>32	4	4 (18.2)	7 (31.8)
Clarithromycin *	0.5	1	18 (81.8)	13 (59.1)
Doxycycline	>256	>16	0 (0)	4 (18.2)
Imipenem	>32	16	1 (4.5)	1 (5)
Linezolid	>256	16	2 (9.1)	10 (45.5)
Minocycline	>256	>8	2 (9.1)	4 (18.2)
Moxifloxacin		8	–	7(31.8)
Tigecycline	4	0.25	10(45.5)	15(68.2)
Tobramycin		16	–	2(9.1)
Trimethoprim/Sulfamethoxazole	>32	>8	3 (13.6)	6 (27.3)
***M. fortuitum*** (10)				
Amikacin	4	2	7 (70)	9 (90)
Cefoxitin		64	–	0(0)
Ceftriaxone	>32	>64	– ^§^	– ^§^
Ciprofloxacin	0.5	≦0.12	6 (60)	5 (50)
Clarithromycin *	32	>16	1 (10)	3 (30)
Doxycycline	>256	>16	2 (20)	2 (20)
Imipenem	>32	16	0 (0)	1 (10)
Linezolid	>256	4	1 (10)	4 (50)
Minocycline	>256	>8	2 (20)	2 (20)
Moxifloxacin		≦0.25	–	5 (50)
Tigecycline	0.12	0.25	3 (30)	7 (70)
Tobramycin		16	–	2 (20)
Trimethoprim/Sulfamethoxazole	>32	4	3 (30)	3 (30)

^§^ Reference range not provided by CLSI M24-A2 guideline. * Result of Clarithromycin was analyzed after 14 days by broth microdilution method. Other drugs were 5 days. – not available.

**Table 3 pathogens-11-00969-t003:** Antimicrobial susceptibility of rapid growing mycobacteria to clarithromycin by broth microdilution method with extended 14 day incubation period.

Organism (No. of Isolates)	Incubation Time (Days)	MIC (μg/mL)				No (%) Isolates	
		Range	50%	90%	Susceptible	Intermediate	Resistant
*M. abscessus*	5	≦0.06–2	0.25	2	21 (95.5)	0 (0)	1 (4.5)
(22)	14	0.12->16	1	>16	13 (59.1)	2 (9.1)	7 (31.8)
*M. fortuitum*	5	0.12->16	4	>16	5 (50)	3 (30)	2 (20)
(10)	14	1->16	>16	>16	3 (30)	1 (10)	6 (60)

**Table 4 pathogens-11-00969-t004:** In vitro susceptibilities of isolates from various sterile body sites of the rapidly growing mycobacteria using the E-test method.

Organism (No. of Iszolates Tested) and Antimicrobial Agents	MIC (μg/mL)			No. (%) Isolates		
	Range	50%	90%	Susceptible	Intermediate	Resistant
***M. abscessus*** (22)						
Amikacin	2->256	>256	>256	5 (23)	2 (9)	15 (68)
Ceftriaxone	>32	>32	>32	– ^§^	– ^§^	– ^§^
Ciprofloxacin	0.25->32	>32	>32	4 (18)	0 (0)	18 (82)
Clarithromycin	0.03–8	0.5	4	18 (82)	2 (9)	2 (9)
Doxycycline	16->256	>256	>256	0 (0)	0 (0)	22 (100)
Imipenem	2->32	>32	>32	1 (5)	1 (5)	20 (91)
Linezolid	0.25->256	>256	>256	2 (9)	0 (0)	20 (91)
Minocycline	0.5->256	>256	>256	2 (9)	0 (0)	20 (91)
Tigecycline	0.03->256	4	>256	– ^§^	– ^§^	– ^§^
Trimethoprim/Sulfamethoxazole	0.064->32	>32	>32	3 (14)	0 (0)	19 (86)
***M. fortuitum*** (10)						
Amikacin	2->256	4	>256	7 (70)	0 (0)	3 (30)
Ceftriaxone	8->32	>32	>32	–	–	–
Ciprofloxacin	0.12->32	0.5	>32	6 (60)	0 (0)	4 (40)
Clarithromycin	1->256	32	>256	1 (10)	2 (20)	7 (70)
Doxycycline	1->256	>256	>256	2 (20)	0 (0)	8 (80)
Imipenem	>32	>32	>32	0 (0)	0 (0)	10 (100)
Linezolid	4->256	>256	>256	1 (10)	2 (20)	7 (70)
Minocycline	0.25->256	>256	>256	2 (20)	1 (10)	7 (70)
Tigecycline	0.03–8	0.12	8	–	–	–
Trimethoprim/Sulfamethoxazole	0.12->32	>32	>32	3 (30)	0 (0)	7 (70)

^§^ Reference range not provided by CLSI M24-A2 guideline. – not available.

**Table 5 pathogens-11-00969-t005:** Antimicrobial susceptibility of *M. abscessus* group and *M. fortuitum* isolates from different time periods in Taiwan using the broth microdilution method.

Antimicrobial Agents	Time Period		
	1997–2001 *	2002–2003 ^#^	2005–2013 (Current Study)
** *M. abscessus* **			
Amikacin	95.5	93.4	86.4
Cefoxitin	3.3	39.9	22.7
Ciprofloxacin	3.3	35.7	31.8
Clarithromycin	79.3	52.7	59.1
Doxycycline	0	3.6	18.2
Imipenem	12.0	28.9	5
Linezolid	31.5	–	45.5
Minocycline	–	50.3	18.2
Moxifloxacin	7.6	–	31.8
Tigecycline	–	–	68.2
Tobramycin	27.2	50.3	9.1
Trimethoprim/Sulfamethoxazole	1.1	–	27.3
** *M. fortuitum* **			
Amikacin	100	96.4	90
Cefoxitin	18.9	64.6	0
Ciprofloxacin	62.3	94.6	50
Clarithromycin	65.2	14.6	30
Doxycycline	13.0	2.7	20
Imipenem	60.9	38.3	10
Linezolid	68.1	–	40
Minocycline	–	53.6	20
Moxifloxacin	66.7	–	50
Tigecycline	–	–	70
Tobramycin	8.7	77.1	20
Trimethoprim/Sulfamethoxazole	49.3	–	30

– not available. * Data from [9]. # Data from [10].

## Data Availability

Not applicable.

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
