# Peer review of "High Rates of Antimicrobial Resistance in Rapidly Growing Mycobacterial Infections in Taiwan"

_pathogens, 2022, doi:10.3390/pathogens11090969_

Round 1
Reviewer 1 Report
In this study, the authors determined and compared MICs of 13 antimicrobial agents using modified broth microdilution and E-test was determined for 32 clinical isolates of RGM, including 22 Mycobacterium abscessus and 10 Mycobacterium fortuitum. They showed that RGM remains susceptible to amikacin, and clarithromycin, but inducible macrolide resistance may occur in over one-third of isolates. This study is a good addition for the appropriate treatment of RGM infections to improve outcome. I have a few comments for further improvement.
1. Delete “This section may be divided by subheadings. It should provide a concise and precise 59 description of the experimental results, their interpretation, as well as the experimental 60 conclusions that can be drawn.” In Introduction.
2. Full name and definition of MIC, MIC50, and MIC90.
3. More details about Broth microdilution test and E test would be helpful in Introduction.
4. Line 109 “without”; line 110 “MIC50 were”; ………many typos. Please check the whole manuscript
5. “The MICs of all test strains against the drug “tigecycline” were low, below 1 μg/mL. In this study, tigecycline showed better activity (68.2% for M. abscessus and 70% for M. fortuitum) than the other antimicrobial agents tested, except for amikacin.” This is confusing writing. And no tigecycline susceptibility data in Table 1.
6. What induce macrolide drug resistance with an extended 14-days incubation period? A brief discussion is needed in section 3.2. What may happen to induce the resistance during the extended incubation?
7. Bar in figure 1 needed.
8. The E-test analysis demonstrated higher levels of antimicrobial resistance among the RGMs compared to broth microdilution in 90% of the isolates. Please give the explanations caused this difference. The authors stated “E-tests will likely result in false resistance due to the higher MICs compared to the broth microdilution tests.” Any approaches can avoid the false E-test results? If just because of the higher MICs values in E-test, is it possible to adjust the susceptibility gating value? Or any principle difference between E test and microdilution?
Author Response
Thank you for your review. We have revised the article according to the reviewer’s suggestions and responded point-by-point as attached below.

Reviewer 2 Report
The manuscript describes a comparison of the antimicrobial susceptibility of rapidly growing mycobacteria (RGM) using the broth microdilution test and E-test for in vitro susceptibility testing, and tries to determine the optimal testing method feasible for routine use in clinical laboratories
Methods are adequate and the results are sound.
Introduction could be improved by describing in more detail previous results.
Following text must be deleted.
lines 59-61 This section may be divided by subheadings. It should provide a concise and precise description of the experimental results, their interpretation, as well as the experimental conclusions that can be drawn
Author Response
Thank you for your review. We have revised the article according to the reviewer’s suggestions and responded point-by-point as attached below. Please see the attachment.

Round 2
Reviewer 1 Report
All comments are addressed.